# FACTORED ACTION SPACES IN DEEP REINFORCEMENT LEARNING

## ABSTRACT

Very large action spaces constitute a critical challenge for deep Reinforcement Learning (RL) algorithms. An existing approach consists in splitting the action space into smaller components and choosing either independently or sequentially actions in each dimension. This approach led to astonishing results for the StarCraft and Dota 2 games, however it remains underexploited and understudied. In this paper, we name this approach Factored Actions Reinforcement Learning (FARL) and study both its theoretical impact and practical use. Notably, we provide a theoretical analysis of FARL on the Proximal Policy Optimization (PPO) and Soft Actor Critic (SAC) algorithms and evaluate these agents in different classes of problems. We show that FARL is a very versatile and efficient approach to combinatorial and continuous control problems.

## 1 INTRODUCTION

In many decision making problems, especially for combinatorial problems, the search space can be extremely large. Learning from scratch in this setting can be hard if not sometimes impossible. Using deep neural networks helps dealing with very large state spaces, but the issue remains when the action space or the horizon required to solve a problem are too large, which is often the case in many real-world settings. Several approaches tackle the problem of long horizon tasks like learning compositional neural programs (Pierrot et al., 2019), hierarchical policies (Levy et al., 2018) or options (Bacon et al., 2017). However, the large action space problem is not as well covered. The main approach consists in factorizing the action space into a Cartesian product of smaller sub-spaces. We call it Factored Actions Reinforcement Learning (FARL). In FARL, the agent must return a sequence of actions at each time step instead of a single one. This approach has been applied successfully to obtain astonishing results in games like StarCraft (Jaderberg et al., 2019), Dota 2 (Berner et al., 2019) or for neural program generation (Li et al., 2020). There was also several attempts to use factored action spaces with DQN, PPO and AlphaZero to solve continuous action problems by discretizing actions and specifying each dimension at a time (Metz et al., 2017; Grill et al., 2020; Tang & Agrawal, 2020). The resulting algorithms outperformed several native continuous action algorithms on MUJOCO benchmarks.

While this approach has been successfully applied in practice, a deeper analysis of the consequences of such a formulation on the RL problem is missing. In this paper, we highlight two different ways to factorize the policy and study their theoretical impact. We discuss the pros and cons of both approaches and illustrate them with practical applications. We extend two state-of-the-art agents PPO and SAC to work with both factorization methods. To highlight the generality of the approach, we apply these algorithms to diverse domains, from large sequential decision problems with discrete actions to challenging continuous control problems and hybrid domains mixing discrete decisions and continuous parameterization of these decisions. We illustrate the method on three benchmarks chosen for the different difficulties they raise and highlight the benefits of using factored actions.

## 2 RELATED WORK

A large part of the reinforcement learning literature covers the long time horizon problem with approaches based on options (Bacon et al., 2017; Vezhnevets et al., 2017), compositionality (Pierrot et al., 2019; 2020) or more generally Hierarchical Reinforcement Learning (Levy et al., 2017; 2018;

Yang et al., 2018; Nachum et al., 2018a;b). However, there has been fewer attempts to deal with large action spaces. In many real life problems, especially in combinatorial or optimisation research standard problems, the number of entities and the size of instances can be very large thus leading to action spaces which may contain thousands of actions. Some prior works have focused on factorizing the action space into binary sub-spaces and using generalized value functions (Pazis & Parr, 2011). A similar approach leveraged Error-Correcting Output Code classifiers (ECOCs) (Dietterich & Bakiri, 1994) to factorize the action space and allow for parallel training of a sub-policy for each action sub-space (Dulac-Arnold et al., 2012). More recently, Dulac-Arnold et al. (2015) proposed to leverage prior information about the actions to embed them into a continuous action space in which the agent can generalize. A concurrent approach is to learn what not to learn (Zahavy et al., 2018). The authors train an action elimination network to eliminate sub-optimal actions, thus reducing the number of possible actions for the RL agent.

In another approach, the action space is factored into a Cartesian product of $n$ discrete sub-action spaces. In Parameterized actions RL, also called Hybrid RL, actions are factored into sequences that correspond to the choice of an action in a discrete action space of size $m$ and then the choice of the intensity of this action in a continuous action space (Hausknecht & Stone, 2015; Masson et al., 2016; Fan et al., 2019; Delalleau et al., 2019). In other problems, the action space exhibits a natural factorization as in Dota 2 or StarCraft. Indeed, one must first choose a macro-action such as selecting a building or a unit and then a sequence of micro-actions such as creating a specific unit, at a specific position. In such a factorization, the autoregressive property is essential, as the selection of an action must be conditioned on the previously selected actions in the sequence. For both games, factorizing the action space and selecting sequences of autoregressive actions instead of single discrete actions has been shown to be crucial (Berner et al., 2019; Vinyals et al., 2019). However neither of these works sufficient highlight this aspect nor propose a proper formalisation.

As far as we know, the only work that establishes a proper FARL framework is Metz et al. (2017) with the model called Sequential DQN (SDQN). They build on existing methods to construct sequential models that have been proposed outside the RL literature. Notably, these models are a natural fit for language modelling Bengio et al. (2003); Sutskever et al. (2014). Metz et al. (2017) extend the DQN algorithm (Mnih et al., 2013) to the sequential setting and present this approach as an alternative way to handle continuous action spaces such as robotic control. Here, we go beyond Q-learning approaches and propose general formulations to extend any actor-critic RL agent to the FARL setting. We illustrate this framework on two examples: we extend both the Proximal Policy Optimization (PPO) (Schulman et al., 2017) and Soft Actor Critic (SAC) algorithms (Haarnoja et al., 2018) to the sequential setting. We also highlight the flexibility and generality of the FARL approach by using it on a broad class of problems. We show results on robotic control MUJOCO benchmarks as in Metz et al. (2017) to demonstrate the relevance of our derivations, and we also successfully apply factored PPO and SAC to parameterized and multi-agent problems.

## 3 FACTORED ACTION SPACES

In this section, we introduce notations for Markov Decision Problems with factored action spaces. We consider a Markov Decision Process (MDP) $(\mathcal{S}, \mathcal{A}, \mathcal{T}, R, \gamma, \rho_0)$ where $\mathcal{S}$ is the state space, $\mathcal{A}$ the action space, $\mathcal{T} : \mathcal{S} \times \mathcal{A} \to \mathcal{S}$ the transition function, $R : \mathcal{S} \times \mathcal{A} \times \mathcal{S} \to \mathbb{R}$ the reward function, $\gamma \leq 1$ is a discount factor and $\rho_0$ is the initial state distribution. We assume that the state space is continuous and that the MDP is fully-observable, thus observations equal states. In this paper, we assume that the action space is factored, thus it might be expressed as a Cartesian product of $n$ discrete action sub-spaces: $\mathcal{A} = \mathcal{A}_1 \times \cdots \times \mathcal{A}_n$ where $\mathcal{A}_i$ is a discrete action space of size $n_i$.

We aim to learn a parameterized stochastic policy $\pi_\theta : \mathcal{A} \times \mathcal{S} \to [0, 1]$, where $\pi(a|s)$ is the probability of choosing action $a$ in state $s$. The objective function to maximise is $\mathcal{J}(\theta) = \mathbb{E}_\tau[\sum_{t=0}^\infty \gamma^t r_t]$ where $\tau$ is a trajectory obtained from $\pi_\theta$ starting from state $s_0 \sim \rho_0$ and $r_t$ is the reward obtained along this trajectory at time $t$. We define the $Q$-value for policy $\pi$, $Q^\pi : \mathcal{S} \times \mathcal{A} \to \mathbb{R}$ as $Q^\pi(s, a) = \mathbb{E}_\tau[\sum_t \gamma^t r_t]$, where $\tau$ is a trajectory obtained from $\pi_\theta$ starting from state $s$ and performing initial action $a$. We define the V-value $V^\pi : \mathcal{S} \to \mathbb{R}$ as $V(s) = \sum_{a \in \mathcal{A}} \pi(a|s) Q^\pi(s, a)$. The policy is factored into a product of $n$ joint distributions to handle the factored action space. We consider two settings.

**Independent Factorization.** A first setting corresponds to problems in which the actions components can be chosen independently from each other, only taking into account the environment state $s$. In this case, we decompose policy $\pi_\theta$ into $n$ policies $\pi_\theta^i : \mathcal{S} \rightarrow \mathcal{A}_i$ such that $\forall (s,a) \in \mathcal{S} \times \mathcal{A}, \ \pi_\theta(a|s) = \prod_{i=1}^n \pi_\theta^i(a^i|s)$ where $a^i$ is the $i^{th}$ component of action $a$. Each sub-policy $i$ returns probabilities over the possible actions $a^i \in \mathcal{A}_i$. In this setting, to sample an action from $\pi_\theta$, we sample in parallel the sub-actions from the sub-distributions $\pi_\theta^i$.

**Autoregressive Factorization.** In this second setting, actions are assumed ordered and dependent. For instance, the choice of action $a^2$ depends on the value of the action $a^1$ that has been chosen by policy $\pi^1$. To account for this, we impose the autoregressive property, i.e. intermediate actions $a^i$ are selected conditionally to previous action choices $a^1, \ldots, a^{i-1}$. More formally, the probability of choosing action $a^i$ is $\pi_\theta^i(a^i|s, a^1, \ldots, a^{i-1})$. As Metz et al. (2017), we introduce sub-state spaces $\mathcal{U}_i = \mathcal{S} \times \mathcal{A}_1 \times \cdots \times \mathcal{A}_i$ where $\mathcal{U}_0 = \mathcal{S}$ and associated sub-states $u_t^i \in \mathcal{U}_i$, which contain the information of the environment state $s_t$ and all the sub-actions that have been selected so far. We decompose the policy $\pi : \mathcal{S} \rightarrow \mathcal{A}$ into $n$ sub-policies $\pi^i : \mathcal{U}_{i-1} \rightarrow \mathcal{A}_i, \ i \in [1, n]$ such that $\forall (s,a) \in \mathcal{S} \times \mathcal{A}, \ \pi_\theta(a_t|s_t) = \prod_{i=1}^n \pi_\theta^i(a_t^i|u_t^{i-1})$. In this setting the sub-actions cannot be sampled in parallel. To sample an action from $\pi_\theta$, we sample each sub-action sequentially, starting from the first and conditioning each sub-policy on the previously sampled actions.

## 4 PROPERTIES OF FACTORED POLICIES

In this section, we discuss the differences between both factorization methods from a theoretical point of view and the impact on their use in practice. We study in particular how the factorization choice affects the expression of the policy entropy and the Kullback-Leibler divergence between two factored policies. Entropies and KL divergences between policies are used in many RL algorithms as regularization terms in order to favor exploration during policy improvement or to prevent policy updates from modifying the current policy too much. When the policy is factored into independent sub-policies, we show that these quantities can easily be computed as the sum of the same quantities computed over the sub-policies. It is not as simple when the policy is autoregressive, but in this case, when computed over actions sampled from the current policy, the sum of the sub-entropies or sub-KL divergences has actually for expected value the global entropy or global KL divergence. The proofs of all propositions are in Appendix B.

### 4.1 SHANNON ENTROPY

The Shannon entropy of a policy is used in several RL algorithms to regularize the policy improvement step or to favor exploration. It is defined as $\mathcal{H}(\pi(.|s)) = - \sum_{a \in \mathcal{A}} \pi(a|s) \log\left(\pi(a|s)\right)$.

**Proposition 1.1** *When the policy is factored into independent policies, its Shannon entropy can be computed as the sum of the Shannon entropies of the sub-policies*:

$$\mathcal{H}(\pi(.|s)) = \sum_{i=1}^n \mathcal{H}(\pi^i(.|s)) \ \text{ with } \ \mathcal{H}(\pi^i(.|s)) = - \sum_{a^j \in \mathcal{A}_i} \pi^i(a^j|s) \log\left(\pi^i(a^j|s)\right). \quad (1)$$

In this setting, the $n$ Shannon entropies can be computed independently in parallel and then summed to obtain the global policy entropy.

**Proposition 1.2** *When the policy is autoregressive, we have:*

$$\mathcal{H}(\pi(.|s)) = \mathbb{E}_{a \sim \pi(.|s)} \left[ \sum_{i=1}^n \mathcal{H}(\pi^i(.|u^{i-1})) \right].$$

This result gives us a simple way to estimate the Shannon entropy of $\pi$. In practice, updates are performed on batches of transitions originating from different states, so the quantity used for regularization is an estimation of $\mathbb{E}_s\left[\mathcal{H}(\pi(.|s))\right]$. Using the above proposition, we know that it equals $\mathbb{E}_s\left[\mathbb{E}_{a \sim \pi(.|s)}\left[\sum_{i=1}^n \mathcal{H}(\pi^i(.|u^{i-1}))\right]\right] = \mathbb{E}_{s,a \sim \pi(.|s)}\left[\sum_{i=1}^n \mathcal{H}(\pi^i(.|u^{i-1}))\right]$.

Therefore, using $\sum_{i=1}^n \mathcal{H}(\pi^i(.|u^i))$ instead of $\mathcal{H}(\pi(.|s))$ for each transition of the batch, we are actually estimating the same quantity. However, it must be noted that the estimation is correct only

if all sequences of actions are sampled according to the current policy $\pi(.|s)$. This does not cause any issue in an *on-policy* context, but when using a replay buffer (and thus transitions obtained with past versions of the policy), the sequences of actions must be resampled with the current policy for the estimation to remain correct.

## 4.2 KULLBACK-LEIBLER DIVERGENCE

The KL-divergence between two policies is also often used in RL, either as a regularization term Schulman et al. (2017) or as a loss function Grill et al. (2020). The KL-divergence between two policies $\pi(.|s)$ and $\mu(.|s)$ is defined as $\text{KL}\left[\pi(.|s)||\mu(.|s)\right] = -\sum_{a\in\mathcal{A}} \pi(a|s)\log\left(\frac{\mu(a|s)}{\pi(a|s)}\right)$.

**Proposition 2.1** *When the two policies are factored into independent sub-policies, their KL-divergence can be computed as the sum of the KL-divergences between the sub-policies*:

$$\text{KL}\left[\pi(.|s)||\mu(.|s)\right] = \sum_{i=1}^{n} \text{KL}\left[\pi^i(.|s)||\mu^i(.|s)\right]. \tag{2}$$

In this setting, the $n$ KL-divergences can be computed independently in parallel and then summed to obtain the final KL-divergence.

**Proposition 2.2** *When the two policies are autoregressive, we have*:

$$\text{KL}\left[\pi(.|s)||\mu(.|s)\right] = \mathbb{E}_{a\sim\pi(.|s)}\left[\sum_{i=1}^{n} \text{KL}\left[\pi^i(.|u^{i-1})||\mu^i(.|u^{i-1})\right]\right]. \tag{3}$$

In this setting, similarly to the Shannon entropy, we use this result to form an estimate where the enriched states $u_i$ are computed from a sequence of actions sampled according to $\pi(.|s)$. Importantly, the sequence of actions used to sequentially compute the sub-distributions must be the same for $\pi$ and $\mu$.

## 5 FACTORED AGENTS

In this section, we highlight the impact of factorizing the action space and of the chosen factorization approach for the policy. In particular, we study two state-of-the-art algorithms, PPO (on-policy) and SAC (off-policy), and show how to adapt them to both factorization settings. We provide guidelines and practical tips to make the factorization work in practice. The relative performance of these algorithms is evaluated in Section 6.

### 5.1 POLICY ARCHITECTURE AND ACTION SAMPLING

We consider a stochastic policy network $\pi_\theta$ taking an environment state $s \in \mathcal{S}$ and returning probabilities over the actions $a \in \mathcal{A}$.

**When the sub-policies are independent distributions**, the policy network is composed of $n$ heads, each taking either directly the state or an embedding of the state computed by a shared neural network between all heads. The $i^{th}$ head returns probabilities over the $i^{th}$ action component $a^i \in \mathcal{A}_i$. The action components are sampled independently and the probability of the resulting action is computed as the product of the components probabilities. As the action components are independent, their sampling can be performed in parallel.

**When the policy is autoregressive**, the policy network is also composed of $n$ heads. However the $i^{th}$ head takes as input not only the state or an embedding but also an encoding of the first $i-1$ selected action components. This encoding can be engineered or computed through a recurrent model. Thus, the $i^{th}$ head returns probabilities over the $i^{th}$ action component $a^i \in \mathcal{A}_i$ conditioned on the choice of the first $i-1$ action components $(a^1, \ldots, a^{i-1})$. Therefore, the action components must be sampled sequentially. As above, the resulting action probability is computed as the product of its components probabilities.

## 5.2 PROXIMAL POLICY OPTIMIZATION

The Proximal Policy Optimization (PPO) algorithm (Schulman et al., 2017) is an on-policy algorithm using a stochastic policy $\pi_\theta$ and a value function $V_\theta$. It replaces the policy gradient by a surrogate loss to constrain the size of policy updates. The policy parameters are updated so as to maximise this loss. An entropy term $\mathcal{H}(\pi_\theta(.|s))$ and a Kullback-Leibler distance between the current policy distribution and its version before the update can be added as regularization terms to further stabilize the training[1].

More formally, policy parameters are updated by gradient descent so as to maximise the following expression:

$$L_\pi(\theta) = \min\left(r(\theta)\hat{A}, \text{clip}(r(\theta), 1-\epsilon, 1+\epsilon)\hat{A}\right) + \mathcal{H}(\pi_\theta(.|s)) - \beta\text{KL}(\pi_{\theta_{old}}(.|s)||\pi_\theta(.|s)),$$

where $\hat{A}$ is an estimate of the advantage and is computed from the target network $V_\theta$ using the Generalized Advantage Estimation method (GAE) (Schulman et al., 2015). The $r(\theta)$ term denotes the policy ratio $r(\theta) = \frac{\pi_\theta(a|s)}{\pi_{\theta_{old}}(a|s)}$ where $\theta_{old}$ are the parameters before the update.

The value network is trained to minimize the mean squared error between its prediction and an estimation of the return. This estimation can be computed from the advantage estimate. More formally, the value function parameters are updated to minimize:

$$L_V(\theta) = \left(V_\theta(s) - (\hat{A} + V_{\theta_{old}}(s))\right)^2.$$

As the value network only depends on states and not on actions, its architecture and update rule are not impacted by the factorization of the action space. It impacts only the policy architecture and its update expression. More precisely: (a) The policy architecture and the way the actions are sampled are modified as explained in Section 5.1; (b) Both $\pi_{\theta_{old}}(a|s)$ and $\pi_\theta(a|s)$ are computed by multiplying the probabilities given by the sub-distributions over the action components; (c) The Shannon entropy and the KL-divergence terms are computed as explained in Sections 4.1 and 4.2.

## 5.3 SOFT ACTOR CRITIC

As we illustrated with PPO, factorizing the action space of an RL algorithm using the $V$-value function is easy. However, things get more complicated when using a $Q$-value function, as actions are involved in the critic. We give a concrete example with the Soft Actor Critic (SAC) (Haarnoja et al., 2018) algorithm. SAC learns a stochastic policy $\pi^*$ maximizing both the returns and the policy entropy. It maximises sums of augmented rewards $r_t^{sac} = r_t + \alpha\mathcal{H}(\pi(.|s_t))$ where $\alpha$ is a temperature parameter controlling the importance of entropy versus reward, and thereby the level of stochasticity of the policy. SAC trains a policy network $\pi_\phi$ for control and a soft Q-function network $Q_\theta$ and relies on a replay buffer $\mathcal{D}$. While the original SAC paper proposes derivations to handle the continuous action setting, the algorithm has been then extended to handle discrete action spaces (Christodoulou, 2019). We use the discrete action derivations as a basis to derive its factored versions. SAC relies on soft policy iteration which alternates between policy evaluation and policy improvement. To extend this algorithm to factored action spaces, we parameterize the policy as explained in Section 5.1.

### CRITIC ARCHITECTURE

A naive parameterization of the critic would use a $Q$-value function taking a state-action pair as input and returning a scalar value. As there are $n_1 \times \cdots \times n_n$ possible actions, training a $Q$-value under this form quickly becomes intractable as the action space grows. To avoid this, we consider a sub $Q$-value for each action dimension.

**In the independent factorization setting**, the action components are selected independently, thus we consider $n$ independent sub $Q$-value functions $Q^i : \mathcal{S} \to \mathcal{A}_i$ which take a state $s$ and return one $Q$-value per possible action in $\mathcal{A}_i$. The $Q$-value function $Q^i$ estimates the average return from

---

[1]The entropy term is not specified in the original paper, but can often be found in available implementations such as in RLlib or Spinning Up.

state $s$ given the sub-action chosen by policy $\pi^i$, regardless of the other sub-actions chosen, i.e. as if it had to assume that the other sub-actions were chosen by the environment. By maintaining these independent $n$ $Q$-value functions, one can perform $n$ independent policy evaluation steps in parallel as well as $n$ independent policy improvement steps in parallel. We also consider $n$ independent temperature parameter $\alpha_i$ to control the entropy versus reward trade-off for each action dimension.

**In the autoregressive setting**, the action chosen by a sub-policy $\pi^i$ is conditioned on the $i-1$ previously chosen action components $(a^1, \ldots, a^{i-1})$. To give a proper meaning to separate $Q$-values in this framework, we reuse the formulation proposed by Metz et al. (2017) and consider two MDPs: the top MDP corresponds to the MDP at hand in which the action space is factored; the bottom MDP is an extended version of the top one. Therefore, between two top MDPs states $s_t$ and $s_{t+1}$, we consider the $n$ intermediate states $u_t^i$. For the value functions to be equal in both MDPs, we set $r = 0$ and $\gamma = 1$ on each intermediate state. In this formulation, the $i^{th}$ $Q$-value function $Q^i : \mathcal{U}_{i-1} \to \mathcal{A}_i$ can be interpreted as an estimate of the average return from a intermediate state $u^i$ given the action choice of policy $\pi^i$. We also introduce an extra $Q$-value function, dubbed "up $Q$-value", $Q_\theta^U : \mathcal{S} \times \mathcal{A} \to \mathbb{R}$ that estimates the return of the policy in the top MDP. We consider $n$ temperature parameters $\alpha_i$, one per action dimension in the bottom MDP, as well as a global temperature parameter $\alpha$ to control the entropy versus reward trade-off in the top MDP.

Below we detail the impact of both factorization approaches on the soft policy evaluation, as this is where most changes are necessary. See Appendix A for the detailed derivations of the policy improvement step and the automatic entropy adjustment step.

SOFT POLICY EVALUATION

To evaluate policy $\pi$, Haarnoja et al. (2018) introduced the soft state value function defined as

$$V(s_t) = \mathbb{E}_{a_t \sim \pi_\phi(.|s_t)} \left[ Q_\theta(s_t, a_t) - \alpha \log \left( \pi_\phi(a_t|s_t) \right) \right]. \tag{4}$$

To compute $V(s_t)$, one must compute an expectation over the distribution $\pi_\phi(.|s_t)$. In the continuous action setting, computing this expectation is replaced by an expectation over a Gaussian using the re-parameterization trick, thus reducing the variance of the estimate. In the discrete action setting, this expectation can be computed directly through an inner product, thus leading to a smaller variance estimate of the soft value, see Appendix A.1 for more details. Note that, while in the continuous action setting the $Q$-value network takes a state and an action and returns a scalar value, its discrete version takes a state and returns a scalar value for each possible action. Finally, the soft $Q$-value network is trained to minimize the soft Bellman residual:

$$J_Q(\theta) = \mathbb{E}_{(s_t, a_t) \sim \mathcal{D}} \left[ \left( Q_\theta(s_t, a_t) - (r_t + \gamma V(s_{t+1}))^2 \right) \right]. \tag{5}$$

**Independent Factorization setting.** When the sub-policies are independent distributions, the $n$ sub $Q$-value functions parameters are updated so as to minimize the soft Bellman residual in Equation (5), where $Q_\theta$ is replaced by $Q_\theta^i$ and the soft value term $V(s_{t+1})$ is replaced by $V^i(s_{t+1})$ computed as

$$V^i(s_t) = \pi_\phi^i(.|s_t)^T \left[ Q_\theta^i(s_t, .) - \alpha_i \log \left( \pi_\phi^i(.|s_t) \right) \right], \tag{6}$$

where $Q_\theta^i(s_t, .)$ stands for the vector of $Q$-values over $\mathcal{A}_i$ given state $s_t$. As these equations are independent from each other, all the updates can be performed in parallel.

**Autoregressive Factorization setting.** When the policy is autoregressive, the first $n-1$ sub $Q$-value functions are updated so as to minimize the soft Bellman residual in Equation (5) where $r = 0$, $\gamma = 1$, $Q_\theta(s_t, a_t)$ is replaced by $Q_\theta^i(u_t^{i-1}, a_t^i)$ and the soft value term $V(s_{t+1})$ is replaced by $V^{i+1}(u_t^i)$. This term is computed using (6) where states $s_t \in \mathcal{S}$ are replaced by sub-states $u_t^{i-1} \in \mathcal{U}_{i-1}$, see Appendix A.2 for more details.

The up $Q$-value function is updated using (4) and (5) where the expectation over the distribution $\pi_\phi(.|s_t)$ is replaced by an estimate using one action sampled from $\pi_\phi(.|s_t)$. The last sub $Q$-value is updated to enforce equality between values in the top and bottom MDPs:

$$J_{Q^n}(\theta) = \mathbb{E}_{(s_t,a_t)\sim\mathcal{D}}\left[\left(Q_\theta^n(u_t^{n-1}, a_t^n) - Q_\theta^U(s_t, a_t)\right)^2\right]. \tag{7}$$

On one side, training the up $Q$-value overcomes the credit assignment problem induced by the zero reward in the bottom MDP. However, as mentioned before, relying on this $Q$-value alone does work in practice. On the other side, training the sub $Q$-values enables computing a better estimate of the expectation over the distribution $\pi_\phi(.|s_t)$.

## 6 Experimental Study

We illustrate the efficiency of FARL on three different use cases. First, we show that factored action spaces are well suited to solve problems in which actions are composed of a discrete part and a continuous part, i.e. parameterized or hybrid action spaces. Second, we show that the autoregressive property can be used in multiagent problems. We use PPO in the challenging Google football environment Kurach et al. (2019). Finally, we evaluate our agents in discretized MuJoCo environments and show that independent factored policies match continuous action ones even in large dimension and can learn even with millions of possible actions. See Appendix C.1 for a summary of the experimental setting and additional experimental details.

### 6.1 Parameterized action spaces

In this section, we highlight the benefits of autoregressive factorization in parameterized action spaces. We use the gym PLATFORM benchmark, introduced in Masson et al. (2016); Bester et al. (2019), in which an agent must solve a platform game by selecting at each step a discrete action among *hop*, *run* or *leap* as well as the continuous intensity of this action. In the original problem formulation, the agent must return one discrete action as well as three continuous actions lying in different ranges. Only the continuous action corresponding to the discrete choice is applied. The environment contains three platforms as well as stochastic enemies to be avoided. The observation is a vector of 9 features. The return is the achievement percentage of the game: 100 corresponds to completion, i.e. the agent reached the extreme left of the platform. By reducing the action space through autoregressive factorization, the three continuous action spaces are transformed into one single discrete space containing $m$ actions corresponding to discrete bins. The agent first chooses a discrete action and then autoregressively selects among the $m$ bins. The selected bin is converted into a continuous value depending on the discrete choice. Thus, there are $3m$ actions. With this transformation, both factored agents reach completion in a few time steps, see Figure 1c. We also observe that Factored PPO (FPPO) is more stable and has a lower inter-seeds variance but reaches a plateau score of 90% while Factored SAC (FSAC) shows more variability but is sometimes able to reach 100%.

### 6.2 Multi-agent benchmark: Google football

In this section, we show that autoregressive factorization also performs well in multi-agent problems. We tested FPPO in the Google football environment where Kurach et al. (2019) have shown good performance using IMPALA. As the authors, we conducted our experiments with the 3 versus 1 Keeper scenario from Football Academy, where we control a team of 3 players who must score against an opponent keeper. Three types of observations exist in this environment. While the authors observed that their algorithms perform better with minimap pictures inputs, we performed well from raw vectors of features as can be seen in Figure 1b. In the original study, a single neural architecture is shared between all players. Each agent receives a vector of general features as well as a one hot code corresponding to its number and chooses a discrete action among 19. Thus, the total number of possible actions is $19^3 = 6859$. Through autoregressivity, we consider only one agent that receives the global features and return a sequence of discrete actions, one per player to be controlled. Therefore, instead of making a choice among 6859 actions, our agent chooses three actions choices among 19 at each time step. We show that through this approach, PPO outperforms IMPALA with fewer computer resources. After $50M$ steps, it reaches an averaged number of goals of $0.91 \pm 0.04$ while IMPALA reaches $0.86 \pm 0.08$ as reported in Kurach et al. (2019). FPPO was trained only for 2 days on 4 CPU cores while IMPALA was trained with 150 CPU cores. The 4 trained agents played 10.000 episodes. Results are averaged over 4 seeds.

To demonstrate the impact of autoregressivity in this setting, we also performed an ablation study in which we kept the same architecture and hyperparameters but removed the autoregressive property. In this case, the PPO policy returns 3 actions, each depending of the environment observation but independent from each other. Figure 1a shows that, without autoregressivity, the ablation can still learn some behavior but quickly plateaus to a poor local optimum while FPPO finds an optimal strategy.

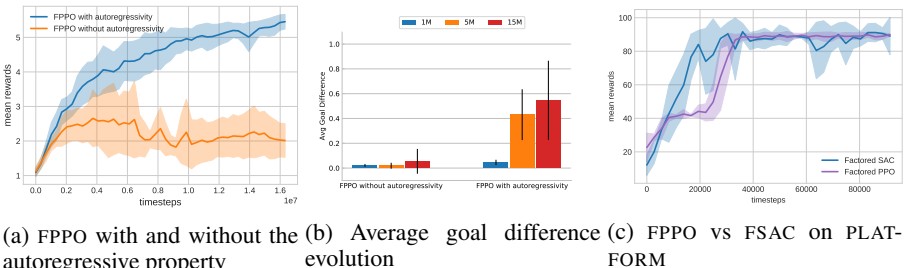

(a) FPPO with and without the autoregressive property  (b) Average goal difference evolution  (c) FPPO vs FSAC on PLATFORM

Figure 1: autoregressive factorization assessment. Figures a) and b) correspond to Factored PPO performance in the Google Football 3 vs 1 keeper environment. Figure c) compares FPPO and FSAC in the gym PLATFORM environment.

## 6.3 MuJoCo benchmarks

Finally, we evaluate our factored agents on four well-documented MuJoCo benchmarks, HalfCheetah-v2, Hopper-v2, Walker2d-v2 and Humanoid-v2. We discretize each of the $n$ continuous action dimensions into $m$ bins, resulting in $m^n$ actions. We use an independent factorization approach as no inter-correlation between action components is needed for those benchmarks for both continuous versions of SAC and PPO. Indeed, these algorithms sample actions from Gaussian distributions with diagonal correlation matrices. We chose $m = 11$ bins for the three benchmarks but observed a low impact of this value on performance. Results are reported in Figure 2. We confirm the results from Tang & Agrawal (2020) for FPPO and demonstrate that FSAC obtains comparable performance to its continuous version despite the discretization. Notably, FSAC performs well on factored Humanoid-v2 which has $\sim 10^{17}$ possible actions, thus demonstrating the scalability of action independent factorization.

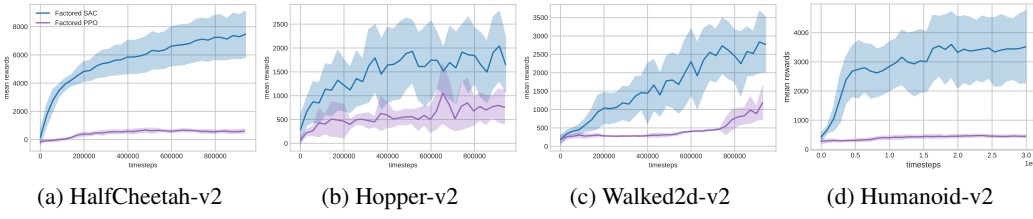

(a) HalfCheetah-v2  (b) Hopper-v2  (c) Walked2d-v2  (d) Humanoid-v2

Figure 2: Independent factorization assessment. FSAC (in blue) vs FPPO (in purple) in factored Mujoco environments. We use independent factorization for both agents.

## 7 Conclusion

Factorizing action spaces leads to impressive results in several RL domains but remains underexploited and understudied. In this work, we studied two factorization methods and highlighted both there theoretical impact on update equations as well as their practical use. We derived practical expression used either to compute or estimate Shannon entropy and Kullback-Leibler divergences. Notably, we showed that action space factorization is well suited for many problems and that these approaches can scale to large number of actions. We used the theoretical study to adapt PPO and SAC, two state-of-the-art agents, to both factorization settings and demonstrated their performance on several benchmarks. We believe that from these derivations and the implementation tips we provided, most of the existing RL agents can be easily adapted to factored action spaces.

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

APPENDIX

## A    FACTORED SOFT ACTOR CRITIC: ADDITIONAL DETAILS

### A.1    EXPECTATION COMPUTATIONS

Several expressions minimised in SAC rely on an expectation over actions sampled according to the policy $\pi_\theta$. In the original version, SAC considers continuous action spaces and assumes a squashed Gaussian policy distribution. Actions are sampled from a Gaussian distribution parameterized by a mean vector $\mu_\theta(s)$ and a diagonal covariance matrix $\sigma_\theta(s)$, both returned by a neural network of weights $\theta$. Then the actions are scaled by a tanh function. More formally, $a = \tanh(e)$ where $e \sim \mathcal{N}(\mu_\theta(s), \sigma_\theta(s))$. This expression can be rewritten as $a = \tanh(\sigma_\theta(s)z + \mu_\theta(s))$ where $z \sim \mathcal{N}(0,1)$. This is called the reparametrization trick; instead of sampling from a distribution which depends on the neural network weights, we sample from a standard normal distribution and apply a linear scaling. This trick enables to rewrite the expectation $\mathbb{E}_{a \sim \pi_\theta(.|s)}$ that depends on $\theta$ as an expectation over the $\mathbb{E}_{z \sim \mathcal{N}(0,1)}$. This trick allows to reduce the variance of this expectation estimate, thus allowing to compute it in practice.

When the action space is discrete, a common choice for the policy distribution is a categorical distribution over actions. In this case, the policy neural network outputs a softmax over the possible actions. This parametrization enables to compute the exact expectation over actions without having to rely on an estimate. Indeed, this expectation can be computed as a simple inner product:

$$\mathbb{E}_{a \sim \pi_\theta(.|s)}[f(a,s)] = \pi_\theta(.|s)^T f(.,s), \tag{8}$$

where $f : \mathcal{S} \times \mathcal{A} \to \mathbb{R}$ is a scalar function and $f(.,s) = [f(a_1,s), \ldots, f(a_n,s)]$ is a vector that contains all the values of $f$ for each possible action $a$. Using this expression reduces the variance of both the actor and critic losses. We use this expression to construct the losses expressions in the factored action settings.

### A.2    SOFT POLICY ITERATION

We demonstrated that in the independent factorization setting, we can use $n$ sub-policies and $n$ sub-$Q$-values, one per action dimension, and update them independently in parallel as if we were solving $n$ parallel MDPs, one per dimension. While this introduces some instability as the reward obtained by one sub-agent depends on the other agents decisions, we observed that this strategy works well in practice. We hypothesize that this performance comes form the fact that modifying the behavior of other agents is slow enough for one agent to improve its own behavior as if the MDP was stationary.

In the autoregressive factorization setting, such a strategy cannot work as agents choices are conditioned on choices of other agents. In this situation, as explained in Section 5.3, we consider two MDPs: the top MDP that corresponds to the MDP at hand in which the action space is factored, and the bottom MDP which is an extended version of the top one. Between two top MDPs states $s_t$ and $s_{t+1}$, we consider the $n$ intermediate states $u_t^i$. We apply to these intermediate states a discount factor that equals 1 and a zero reward so as to ensure value equality on the shared states between both MDPs. In this case, the sub-$Q$-value functions estimate the $Q$-values on bottom states: $Q_\theta^i$ estimates $Q$-values for states $u_t^{i-1}$. We also consider a $Q$-value function $Q_\theta^U$ estimating returns on top states. Both $Q$-values types are complementary: the top $Q$-value suffers from the large number of possible actions. The sub-$Q$-values do not have this issue as they rely on smaller number of actions, i.e. the number of actions per dimension, but suffer from the credit assignment problem induced by the extra zero rewards. We observed that training the top $Q$-value alone does not work, however adding these sub-$Q$-values enables learning. On one side, we train the up $Q$-value so as to minimize soft Bellman residuals in the top MDP, see (5) and on the other we train all sub-$Q$-values, except the last one, to minimize soft Bellman residuals in the bottom MDP:

$$J_Q^i(\theta) = \mathbb{E}_{(s_t,a_t) \sim \mathcal{D}}\left[\left(Q_\theta^i(u_t^{i-1}, a_t^i) - (r_t + \gamma V^{i+1}(u_t^i))^2\right], \ i < n, \tag{9}$$

where $V^i(u_{i-1}^t) = \mathbb{E}_{a_t \sim \pi_\phi(.|s_t)}\left[Q_\theta^i(u_t^{i-1}, a_t^i) - \alpha \log(\pi_\phi^i(u_t^{i-1}|s_t))\right]$. Finally, the last sub $Q$-value is updated to enforce equality between values in the top and bottom MDPs:

$$J_{Q^n}(\theta) = \mathbb{E}_{(s_t, a_t) \sim \mathcal{D}} \left[ \left( Q_\theta^n(u_t^{n-1}, a_t^n) - Q_\theta^U(s_t, a_t) \right)^2 \right]. \tag{10}$$

Note that in all previous expressions, the expectation over actions has been sampled through an inner product as explained in (8).

### A.3 Soft Policy improvement

The policy improvement step updates policy $\pi_\phi$ by minimizing the cost function:

$$J_\pi(\phi) = \mathbb{E}_{a_t \sim \pi_\phi(.|s_t)} \left[ \alpha \log \pi_\phi(a_t|s_t) - Q_\theta(s_t, a_t) \right].$$

As before, the expectation approximated by a Monte Carlo estimate in the continuous action setting is replaced by the true expectation in the discrete action setting.

When the sub-policies are independent distributions, the $n$ sub-policies parameters are updated independently so as to minimize:

$$J_{\pi^i}(\phi) = \pi_\phi^i(.|s_t)^T \left[ \alpha_i \log \pi_\phi^i(.|s_t) - Q_\theta^i(s_t, .) \right], \quad 1 \le i \le n. \tag{11}$$

When the policy is autoregressive, the $n$ sub policies are updated so as to minimize the same loss in which states $s_t \in \mathcal{S}$ are replaced by sub-states $u_t^{i-1} \in \mathcal{U}_{i-1}$. Each sub-policy is updated from its corresponding sub $Q$-value function.

### A.4 Automating entropy adjustment

The entropy adjustment coefficient $\alpha$ can either be fixed as an hyper-parameter or updated online so as to ensure that the policy entropy does go below an entropy target value $\bar{\mathcal{H}}$. Haarnoja et al. (2018) show that $\alpha$ can be updated at each iteration so as to minimize

$$J(\alpha) = \mathbb{E}_{a \sim \pi_\theta(.|s)} \left[ -\alpha \left( \log \pi_\theta(a|s) + \bar{\mathcal{H}} \right) \right]. \tag{12}$$

In this work, we express the entropy target as a fraction $\beta \in [0, 1]$ of the maximum entropy, i.e. the entropy of the uniform distribution over $\mathcal{A}$ that we note $\mathcal{H}_u$. We give some expressions of $\mathcal{H}_u$ corresponding to different action spaces in Table 1.

Table 1: Uniform distributions entropies.

| Action space | Entropy of uniform distribution |
|---|---|
| $\mathcal{A} = \{1, \ldots, n\}$ | $\mathcal{H}_u = \log n$ |
| $\mathcal{A} = [-1, \ 1]^n$ | $\mathcal{H}_u = n \log 2$ |
| $\mathcal{A} = \mathcal{A}_1 \times \cdots \times \mathcal{A}_n$ | $\mathcal{H}_u = \sum_{i=1}^n \log n_i$ |

When the factorization is independent, we simply minimize $n$ independent losses, one for each sub-entropy coefficient $\alpha_i$:

$$J(\alpha_i) = \pi_\theta^i(.|s)^T \left[ -\alpha_i \left( \log \pi_\theta^i(.|s) + \bar{\mathcal{H}}_i \right) \right], \tag{13}$$

where $\bar{\mathcal{H}}_i = \beta \log(n_i)$.

When the factorization is autoregressive, we minimize the same $n$ losses where states $s_t$ are simply replaced by sub-states $u_t^{i-1}$. We also optimize the entropy coefficient for the top MDP $\alpha$ by minimizing expression (12) where the entropy target is computed as $\bar{\mathcal{H}} = \beta \sum_{i=1}^n \log(n_i)$. We found in practice that the choice of parameter $\beta$ has the greatest impact on performance. We show an example on the Humanoid-v2 benchmark in Section C.2.

# B    PROPERTIES OF FACTORED POLICIES - PROOFS

## B.1    SHANNON ENTROPY

PROOF OF PROPOSITION 1.1

When the policy $\pi$ is factored into independent sub-policies, all partial actions $a^i$ depend only on the state $s$. In this case, the entropy of the policy is by definition the joint entropy of the random variables associated to each partial action. Furthermore, this joint entropy is equal to the sum of the entropies of each sub-policy, due to the additivity of entropy for independent random variables:

$$
\begin{aligned}
\mathcal{H}(\pi(.|s)) &= - \sum_{a=(a^1,\ldots,a^n)\in\mathcal{A}_1\times\cdots\times\mathcal{A}_n} \pi(a|s)\log\left(\pi(a|s)\right) \\
&= - \sum_{a^1,\ldots,a^n\in\mathcal{A}_1\times\cdots\times\mathcal{A}_n} \left(\prod_{i=1}^n \pi^i(a^i|s)\log\left(\prod_{i=1}^n \pi^i(a^i|s)\right)\right) \\
&= \sum_{i=1}^n \mathcal{H}(\pi^i(.|s)).
\end{aligned}
$$

Remark: see (Gray, 2011) for a proof of the additivity of the Shannon entropy for independent random variables. In the case of two discrete independent random variables $X$ and $Y$, it can be demonstrated as follows:

$$
\begin{aligned}
\mathcal{H}(X,Y) &= - \sum_{x,y} P(x,y)\log\left(P(x,y)\right) \\
&= - \sum_{x,y} P(x)P(y)\log\left(P(x)\right) - \sum_{x,y} P(x)P(y)\log\left(P(y)\right) \\
&= - \sum_{x} P(x)\log\left(P(x)\right) - \sum_{y} P(y)\log\left(P(y)\right) \\
&= \mathcal{H}(X) + \mathcal{H}(Y).
\end{aligned}
$$

PROOF OF PROPOSITION 1.2

When the discrete random variables $X$ and $Y$ are not necessarily independent, the more general equation is that the joint entropy is equal to the sum of the entropy of $X$ and the conditional entropy $\mathcal{H}(Y|X) = - \sum_{x,y} P(x,y)\log\left(\frac{P(x,y)}{P(x)}\right)$:

$$
\mathcal{H}(X,Y) = \mathcal{H}(X) + \mathcal{H}(Y|X).
$$

It can be expressed as an expected value:

$$
\begin{aligned}
\mathcal{H}(X,Y) &= \mathcal{H}(X) - \sum_{x,y} P(x)\frac{P(x,y)}{P(x)}\log\left(\frac{P(x,y)}{P(x)}\right) \\
&= \mathcal{H}(X) - \sum_{x} P(x)\sum_{y} P(y|x)\log\left(P(y|x)\right) \\
&= \mathcal{H}(X) + \mathbb{E}_x\left[\mathcal{H}(Y|x)\right].
\end{aligned}
$$

Or for three random variables: $\mathcal{H}(X,Y,Z) = \mathcal{H}(X) + \mathbb{E}_x\left[\mathcal{H}(Y|x)\right] + \mathbb{E}_{x,y}\left[\mathcal{H}(Z|x,y)\right]$, and this can be generalized further. Applying it to an autoregressive policy $\pi(a|s) = \prod_{i=1}^n \pi^i(a^i|u^{i-1})$, it yields the following equation:

$$
\mathcal{H}(\pi(.|s)) = \mathcal{H}(\pi^1(.|s)) + \mathbb{E}_{u^1}\left[\mathcal{H}(\pi^2(.|u^1))\right] + \mathbb{E}_{u^2}\left[\mathcal{H}(\pi^3(.|u^2))\right] + \cdots + \mathbb{E}_{u^{n-1}}\left[\mathcal{H}(\pi^n(.|u^{n-1}))\right],
$$

which can be written:

$$\mathcal{H}(\pi(.|s)) = \mathbb{E}_{a \sim \pi(.|s)} \left[ \sum_{i=1}^{n} \mathcal{H}(\pi^i(.|u^{i-1})) \right].$$

## B.2 KULLBACK-LEIBLER DIVERGENCE

PROOF OF PROPOSITION 2.1

Let us consider 4 independent discrete random variables $X$, $Y$, $X'$ and $Y'$, $X$ and $X'$ having same support and respectively probability mass functions $p$ and $p'$, and $Y$ and $Y'$ having same support and respectively probability mass functions $q$ and $q'$. We denote by $\mathcal{H}(X,Y|X',Y')$ the cross-entropy between the joint distributions of $X$, $Y$ and $X'$, $Y'$:

$$\mathcal{H}(X,Y|X',Y') = -\sum_{x,y} p(x)q(y) \log p'(x)q'(y).$$

We have:

$$\begin{aligned}
\mathcal{H}(X,Y||X',Y') &= -\sum_{x,y} p(x)q(y) \log p'(x) - \sum_{x,y} p(x)q(y) \log q'(y) \\
&= -\sum_{x} p(x) \log p'(x) - \sum_{y} q(y) \log q'(y).
\end{aligned}$$

Therefore:

$$\mathcal{H}(X,Y||X',Y') = \mathcal{H}(X||X') + \mathcal{H}(Y||Y').$$

The Kullback-Leibler divergence, cross-entropy and entropy are linked by the following formula:

$$\mathrm{KL}[Z_1||Z_2] = \mathcal{H}(Z_1||Z_2) - \mathcal{H}(Z_1).$$

Using the additivity of entropy for independent random variables:

$$\begin{aligned}
\mathrm{KL}[X,Y||X',Y'] &= \mathcal{H}(X,Y||X',Y') - \mathcal{H}(X,Y) \\
&= \mathcal{H}(X||X') + \mathcal{H}(Y||Y') - \mathcal{H}(X) - \mathcal{H}(Y) \\
&= \mathcal{H}(X||X') + \mathcal{H}(Y||Y') - \mathcal{H}(X) - \mathcal{H}(Y) \\
&= \mathrm{KL}[X||X'] + \mathrm{KL}[Y||Y'].
\end{aligned}$$

This can be generalized to joint distributions of more than 2 independent discrete random variables. Applied to the context of policies in factored action spaces, and assuming that $\pi$ and $\mu$ are two policies such that $\forall (s,a) \in \mathcal{S} \times \mathcal{A}$, $\pi^i(a|s) = \prod_{i=1}^{n} \pi^i(a^i|s)$ and $\mu^i(a|s) = \prod_{i=1}^{n} \mu^i(a^i|s)$, it results in

$$\mathrm{KL}\left[\pi(.|s)||\mu(.|s)\right] = \sum_{i=1}^{n} \mathrm{KL}\left[\pi^i(.|s)||\mu^i(.|s)\right].$$

PROOF OF PROPOSITION 2.2

As in the proof of the previous proposition, we consider 4 discrete random variables $X$, $X'$, $Y$ and $Y'$. $X$ and $X'$ have the same support and are independent, $Y$ and $Y'$ have the same support and are independent, but this time $X$ and $Y$ are not independent, and $X'$ and $Y'$ are not independent. We respectively denote their joint probability mass functions by $p$ and $q$. Without ambiguity, we also denote by $p$ and $q$ the marginalizations over $y$: $p(x) = \sum_y p(x,y)$ and $q(y) = \sum_y q(x,y)$.

Let us consider the cross-entropy between the joint probability distributions of $X, Y$ and $X', Y'$ :

$$
\begin{aligned}
\mathcal{H}(X, Y || X', Y') &= -\sum_{x,y} p(x, y) \log q(x, y) \\
&= -\sum_{x,y} p(x, y) \log q(x, y) + \sum_{x,y} p(x, y) \log q(x) - \sum_{x,y} p(x, y) \log q(x) \\
&= -\sum_{x,y} p(x, y) \log \frac{q(x, y)}{q(x)} - \sum_{x} p(x) \log q(x) \\
&= -\sum_{x,y} p(x) \frac{p(x, y)}{p(x)} \log \frac{q(x, y)}{q(x)} - \sum_{x} p(x) \log q(x) \\
&= -\sum_{x} p(x) \sum_{y} p(y|x) \log q(y|x) - \sum_{x} p(x) \log q(x) \\
&= \mathcal{H}(X || X') + \mathbb{E}_{x \sim p} \mathcal{H}(Y|x || Y'|x).
\end{aligned}
$$

Using the equality $\mathrm{KL}[Z_1 || Z_2] = \mathcal{H}(Z_1 || Z_2) - \mathcal{H}(Z_1)$, and the equality derived in the proof of Proposition 1.2, we get:

$$
\begin{aligned}
\mathrm{KL}[X, Y || X', Y'] &= \mathcal{H}(X || X') + \mathbb{E}_{x \sim p}\left[\mathcal{H}(Y|x || Y'|x)\right] - \mathcal{H}(X, Y) \\
&= \mathcal{H}(X || X') - \mathcal{H}(X) + \mathbb{E}_{x \sim p}\left[\mathcal{H}(Y|x || Y'|x)\right] - \mathbb{E}_{x \sim p}\left[\mathcal{H}(Y|x)\right] \\
&= \mathrm{KL}\left[X || X'\right] + \mathbb{E}_{x \sim p}\left[\mathrm{KL}(Y|x || Y'|x)\right].
\end{aligned}
$$

Similarly, for random variables $X, Y, Z$ and $X', Y', Z'$, we obtain

$$
\mathrm{KL}[X, Y, Z || X', Y', Z'] = \mathrm{KL}\left[X || X'\right] + \mathbb{E}_{x \sim p}\left[\mathrm{KL}(Y|x || Y'|x)\right] + \mathbb{E}_{(x,y) \sim p}\left[\mathrm{KL}(Z|x, y || Z'|x, y)\right].
$$

Again, the equation can be generalized to joint distributions of $n$ discrete random variables. In the context of autoregressive policies, assuming that $\pi$ and $\mu$ are two policies such that $\forall (s, a) \in \mathcal{S} \times \mathcal{A}$, $\pi^i(a|s) = \prod_{i=1}^{n} \pi^i(a^i|u^{i-1})$ and $\mu^i(a|s) = \prod_{i=1}^{n} \mu^i(a^i|u^{i-1})$, it results in

$$
\mathrm{KL}\left[\pi(.|s) || \mu(.|s)\right] = \mathbb{E}_{a \sim \pi(.|s)}\left[\sum_{i=1}^{n} \mathrm{KL}\left[\pi^i(.|u^{i-1}) || \mu^i(.|u^{i-1})\right]\right].
$$

# C    ADDITIONAL EXPERIMENTAL RESULTS

## C.1    EXPERIMENTAL SETTING SUMMARY

Table 2: Experimental setting summary

| Environment | Nb of actions dimensions | Nb of actions per dimension | Total nb of actions | Factorization |
|---|---|---|---|---|
| Google Football | 3 | 19 | 6859 | Autoregressive |
| Platform | 2 | (3, 21) | 63 | Autoregressive |
| Humanoid-v2 | 17 | 11 | 5e17 | Independent |
| HalfCheetah-v2 | 6 | 11 | 1.7e6 | Independent |
| Walker2d-v2 | 6 | 11 | 1.7e6 | Independent |
| Hopper-v2 | 3 | 11 | 1.8e5 | Independent |

## C.2    IMPACT OF ENTROPY TARGET STUDY

We study in Figure 3 the impact of coefficient $\beta$ that defines the entropy target on the agent performance. This parameter defines the balance between exploration and exploitation. When its value tends to 1, $\alpha$ is tuned so as to maintain a policy entropy near the maximum entropy while when its value tends to 0, the policy becomes almost deterministic. We observe that in Humanoid-v2 this value must remain small so as to ensure convergence.

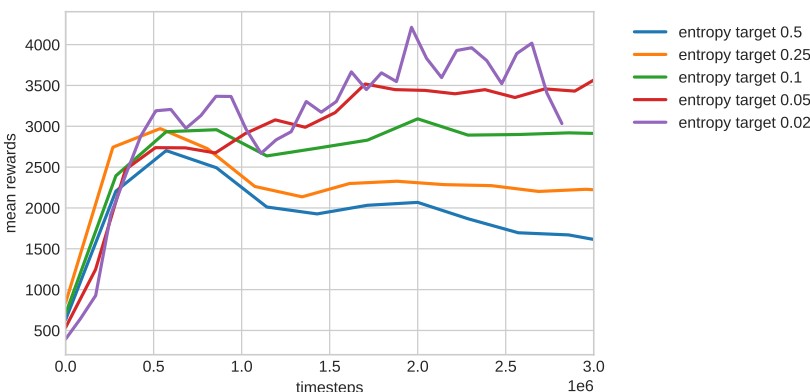

Figure 3: Study on the impact of the entropy target in FSAC on the Humanoid-v2 environment. Each run has been averaged over 4 seeds. In this example, all dimensions share the same entropy target $\bar{\mathcal{H}}_i = \beta \log 17$. Each run represented corresponds to a different value of parameter $\beta$.

## C.3 HYPERPARAMETERS

Table 3: Hyper-parameters for the MUJOCO environments

(a) FPPO

| Parameter | Value |
|---|---|
| Optimizer | Adam |
| Learning rate | $3.10^{-4}$ |
| Discount factor | 0.99 |
| Number of bins per dimension | 11 |
| Clipping epsilon | 0.2 |
| KL coefficient | 0.2 |
| Lambda GAE | 0.95 |
| Hidden layers size | 256/256 |
| Activations | ReLU |
| Minibatch size | 128 |
| Number of epochs | 20 |
| Memory size | 2048 |

(b) FSAC

| Parameter | Value |
|---|---|
| Optimizer | Adam |
| Learning rate | $3.10^{-4}$ |
| Discount factor | 0.99 |
| Number of bins per dimension | 11 |
| Replay buffer size | $10^6$ |
| Hidden layers size | 256/256 |
| Activations | ReLU |
| Minibatch size | 256 |
| Target smoothing coefficient | 0.005 |
| Entropy target $\beta$ | 0.05, 0.1, 0.5[a] |

[a]respectively $\beta = 0.05$ in Humanoid-v2, $\beta = 0.1$ in HalfCheetah-v2 and Walker2d-v2 and $\beta = 0.5$ in Hopper-v2

Table 4: Hyper-parameters for the PLATFORM environment

(a) FPPO

| Parameter | Value |
|---|---|
| Optimizer | Adam |
| Learning rate | $5.10^{-5}$ |
| Discount factor | 0.99 |
| Number of bins per dimension | 21 |
| Clipping epsilon | 0.2 |
| KL coefficient | 0.2 |
| Lambda GAE | 1 |
| Hidden layers size | 128/128 |
| Activations | ReLU |
| Minibatch size | 64 |
| Number of epochs | 20 |
| Memory size | 512 |

(b) FSAC

| Parameter | Value |
|---|---|
| Optimizer | Adam |
| Learning rate | $3.10^{-4}$ |
| Discount factor | 0.99 |
| Number of bins per dimension | 21 |
| Replay buffer size | $10^6$ |
| Hidden layers size | 128/128 |
| Activations | ReLU |
| Minibatch size | 256 |
| Target smoothing coefficient | 0.01 |
| Entropy target $\beta$ | 0.5 |

Table 5: FPPO hyper-parameters for the Google Football environment

| Parameter | Value |
|---|---|
| Optimizer | Adam |
| Learning rate | $3.10^{-4}$ |
| Discount factor | 0.99 |
| Clipping epsilon | 0.08 |
| KL coefficient | 0.2 |
| Lambda GAE | 0.95 |
| Hidden layers size | 256/256 |
| Activations | ReLU |
| Minibatch size | 375 |
| Number of epochs | 2 |
| Memory size | 3000 |

