# OpenReview forum: "Factored Action Spaces in Deep Reinforcement Learning"
_ICLR.cc/2021/Conference — Reject_

### Official Review · AnonReviewer3 · 2020-10-25
**Good direction; lack of clarity regarding final message**

**Rating:** 5
**Confidence:** 4

**Review:**

### Summary
The paper presents a study of factored action spaces for RL problems, and two basic forms of this,
i.e., independent factorization (IF) and autoregressive factorization (AF).
Note: these acronyms are my own. Building on these two types of factorization, the paper presents:
(a) derivations of the entropy and KL-divergence for these policy factorizations;
(b) PPO and SCA adaptations for IF + AF; and
(c) three categories of experimental results.
The experiments compare, in separate scenarios:  (i) AF-PPO vs AF-SAC; (ii) AF-PPO vs IMAPLA and non-autoregressive F-PPO;
(iii) IF-SAC vs IF-PPO.
Overall, the experiments show some of the possible potential of factored action spaces with PPO and SAC.

### Strengths
- I believe that factored action space are an under-explored area of research, and
  understanding the benefits and limitations of these factorizatios is an important area,
  as well as understanding how popular algorithms such as PPO and SCA can be adapted to use these factorizations.
- The adaptations of PPO and SCA and the related derivations, while not entirely surprising, are useful.
- The experiments show some of the potential of factored action spaces, both IF and AF.
  The experimental parameters appear to be well documented, c.f., Appendix C.3.

### Weaknesses
- There are no guidelines or experiments to help determine when the factored approaches should be used,
  i.e., when do they do better and when they do worse.  When is it harmful to use a factored approach?
  Similarly, when should IF vs AF be used?
- Related to the point above, the experiments generally reflect the idea that the implementations can be made to work on some examples, rather than understanding where these factored actions spaces sit in the broader universe of algorithmic choices. What motivated the given choices of three benchmark problems, and the specific  (limited) comparisons shown for each benchmark?  It is difficult to fully understand the "take home message"  of each of the experiments.
- It would be good to understanding more about the relationship with possible continuous action derivations

### Recommendation
I am on the fence regarding this paper.
Exploring and understanding factored action spaces is important, and the paper presents a reasonable step
in this direction, and will therefore provide a foundation for further work.
However there is little provided in the way of insights as to understanding the types of situations
where factored representations should be used, and how this plays our for PPO and SCA.
If pressed, I currently lean slightly in favor.

### Questions
- What is the impact of the action ordering for the autoregressive factorization?
- It is stated that there is no inter-correlation between action components for the MuJoCo benefits.
  What evidence do you have for this?  The statement is unintuitive to those working on continuous control problems.
  Why can't there be multimodal areas of the state space where the action decisions need to be coordinated?

### Additional Feedback
- section 5.3 Critic Architecture
  "As there are n1 ... nn possible actions".  The notation is flawed, given that n is used to represent
  both the number of dimensions and the discretization. Similarly, later it is stated:
  "these independent n Q-value functions" is confusing, given that each of N dimensions presumably each need n Q-value
  functions in the discrete action setting.
- section 5.3 "we reuse the formulation proposed by Metz et al. (2017)
  and "The top MDP corresponds to the MDP at hand in which the action space is factored"
  Should make it clear that this refers to Figure 1 of that paper.
  And I would refer to the top MDP as being the original unfactored MDP.
- section 5.3 Soft Policy Evaluation "computed directly through an inner product"
  Could in include a forward reference to eqn (6)?
- equation (5): missing right bracket
- section 6.1 "both factored agents reach completion in a few time steps, see Figure 1c"
  Figure 1c shows 20k - 40k timesteps, so while that is small, it is strange to call that "a few time steps"
- section 6.2: "our agent chooses three actions [sic] choices among 19 at each time step."
  Why not explicitly state 3x19 = 57 choices at each time step (3 successive choices of 19 each).
- The relationship to (Tang and Agrawal 2020) could be better explained.
- A diagrammatic abstraction of the different forms of factorization could be very useful to the reader.

Update:
  After reading the other reviews and the responses, I have changed my score to 5: marginally below acceptance, due to the framing and related work issues, as discussed by R2 and R4.  There is potential here, but it will benefit from strong revisions.

---

> ### Author Response · Authors · 2020-11-24
> **Response to reviewer 3**
>
> As reviewer 4, the reviewer asks about the impact of the action ordering for the autoregressive factorization. See our answer to Reviewer 4.
>
> The reviewer also asks for evidence that there is no inter-correlation between action components for the MuJoCo benefits. A first reason for our factorization choice is that stochastic continuous agents such as SAC parameterize a diagonal covariance matrix assuming non diagonal coefficients to always equal zero. A second reason is that we do not think that there would be any meaningful order for the actions in the MuJoCo benchmarks. Finally, we had great difficulty to obtain decent results on the MuJoCo benchmarks with auto-regressive factorization (the algorithm became very sensitive to hyper-parameters) while the independent factorization worked immediately with no need to tune any hyper-parameter.

---

### Official Review · AnonReviewer2 · 2020-10-27

**Rating:** 3
**Confidence:** 5

**Review:**

##########################################################################
Summary:

The paper studies policy optimization in multidimensional action spaces. They consider atomic factorization of the action space (i.e. action space is factored into sub-action spaces, one per action dimension). In this setting, the authors consider two well-known policy representation techniques: 1) independent sub-policies (diagonal-covariance policies over the sub-action spaces) and 2) sequential/autoregressive policies (an ordering of the sub-action spaces is assumed a priori and the sub-policies receive as input the state and the selected sub-actions for the preceding sub-action spaces). The authors develop methods based on these two factorization techniques for discrete versions of PPO and SAC (called FPPO and FSAC) and evaluate them on Gym Platform, Google Football, and discretized MuJoCo tasks.

##########################################################################
Reasons for score:

I believe this paper fails to position itself appropriately with respect to the literature. This makes it difficult to assess what the main contributions are and whether sufficient new methodology is proposed. Additionally, the paper misses on several fundamental experiments for the results to be convincing. Also, the paper does not set a clear agenda for what would be interesting to see in the results; Is scalability to large action spaces being investigated (in which case, comparison with the non-factored baselines of PPO and SAC should be included)? Is it the improvements of the discrete versions over the continuous versions of the methods that are interesting (in which case, the results for the continuous versions on the MuJoCo tasks should be reported)?

##########################################################################
Pros:

- Factorizing action spaces is an interesting approach for scaling to high-dimensional action spaces. Also, they have been shown in several recent studies to enable discrete-action methods to outperform numerous continuous-control methods. So studying them more extensively is useful.

- The paper nicely outlines how PPO and SAC should be configured to work with independent and autoregressive policies under the atomic factorization of the action space.

##########################################################################
Cons:

1) The paper generally does not position itself appropriately with respect to the literature. Below is my overview of the paper with additional related works (some are suggestions that would enhance the paper):

- Branching Q-learning (BQL) in Ref. [I] is similar (if not the same) to the independent critic of FSAC. Using this approach they scale to domains with 33^17 discrete actions.

- Autoregressive critic of FSAC is the same as that in Metz et al. (2017) (also stated in the paper I believe).

- Independent versions of numerous PG methods have been previously developed and studied extensively by Tang and Agrawal (2020). Also, the original TRPO paper was used with independent discrete sub-action policies in Atari tasks (if I remember correctly). Therefore, there is nothing new about FPPO-independent (also stated in the paper I believe).

- Other methods for learning the independent critic in FSAC are also possible but are not discussed in the paper (see e.g. Ref. [II]). Moreover, Ref. [II] uses independent policies with A2C.

- When independent policies are used in PG, a better baseline can be used which provably reduces variance [III]. Discussion of such a baseline would be very useful in this paper which somewhat serves to summarize deep RL in factored action spaces.

- Ref. [IV] combines autoregressive and independent proposal policies for approximate Q-maximization in Q-learning. In this way, they scale to very high-dimensional action spaces. This work is able to handle hybrid action spaces without discretization. A discussion and positioning with respect to this paper could be valuable.


2) Experiments fall very short in my opinion. Below are some of the experiments that I think are necessary to include:

- Figure 1a,b: Why not run FSAC?

- Figure 1c: I believe the autoregressive FPPO and FSAC are reported in Platform. But why not independent? Why not report the standard (non-factored) baselines of PPO and SAC?

- Figure 2: Compare against the continuous versions of SAC and PPO on MuJoCo.

- Figure 2: Why not evaluate autoregressive policies in MuJoCo? Metz et al. did that with DQN.


3) No new factorization schemes are explored in this paper. For instance, Ref. [V] explores a mixture of independent and sequential action-value function representations. A discussion of the spectrum of other possible factorizations would be interesting.


[I] Action Branching Architectures for Deep Reinforcement Learning, AAAI 2018.
[II] Learning to Factor Policies and Action-Value Functions: Factored Action Space Representations for Deep Reinforcement Learning, arXiv 2017.
[III] Variance Reduction for Policy Gradient with Action-Dependent Factorized Baselines, ICLR 2018.
[IV] Q-Learning in enormous action spaces via amortized approximate maximization, arXiv 2020.
[V] Inferring DQN structure for high-dimensional continuous control, ICML 2020.

##########################################################################
Questions during the rebuttal period:

1) In light of new related works together with those included in the paper, I believe the novelty of the paper currently is in developing the policy optimization updates for autoregressive policies. The rest (independent policy optimization updates, independent critic, and autoregressive critic) are not novel as far as my assessment goes at this point. Is this correct?

2) Does "without autoregressive" (in Figure 1) imply "independent"? Or does it refer to the standard (non-factored) versions of PPO and SAC? If not, I need clarification on what it exactly means.

3) This paper considered stochastic policies. I'm curious about any potential use for extending it to deterministic policies; e.g. could it be useful to use an autoregressive policy network with DDPG?

4) I think that the manual ordering of the sub-action spaces in an auto-regressive policy could introduce bias (i.e. I think there could be a setting that some orderings would make it impossible to learn the optimal policy). Can you comment on this?

##########################################################################
Minor comments:

- The title is too broad in my opinion. Unless the paper also incorporates analysis and results for purely action-value methods (e.g. DQN-based agents such as Sequential and Branching Q-learning), the paper should specify that policy optimization methods are of focus. Also, stating atomic factorization in multidimensional action spaces instead of only "factored" could clarify the kind of factorization that is the subject of this paper.

- "FPPO was trained only for 2 days on 4 CPU cores while IMPALA was trained with 150 CPU cores.":
This is not very useful without providing training-time of IMPALA.

- The word action is frequently used instead of action dimension and sub-action.

- "With this transformation, both factored agents reach completion in a few time steps, see Figure 1c":
Saying "a few time steps" for a plot that goes to 90k time steps is somewhat strange.

- This work claims to develop techniques for dealing with hybrid action spaces. But in the experiments, this is not demonstrated. In fact, the action space in Gym Platform features discrete and continuous sub-action spaces, but they are discretized. As such, I think this paper should state dealing with discrete action spaces or continuous ones via discretization.

---

> ### Author Response · Authors · 2020-11-24
> **Response to reviewer 2**
>
> We warmly thank the reviewer 2 for the thorough review and all the additional references we missed. These references will strengthen our future version of the paper.
>
> The reviewer asked:
>
> 1) In light of new related works together with those included in the paper, I believe the novelty of the paper currently is in developing the policy optimization updates for autoregressive policies. The rest (independent policy optimization updates, independent critic, and autoregressive critic) are not novel as far as my assessment goes at this point. Is this correct?
>
> Yes it is.
>
> 2) Does "without autoregressive" (in Figure 1) imply "independent"? Or does it refer to the standard (non-factored) versions of PPO and SAC? If not, I need clarification on what it exactly means.
>
> In the Google Football environment, “without auto-regressivity” indeed means independent.
>
> 3) This paper considered stochastic policies. I'm curious about any potential use for extending it to deterministic policies; e.g. could it be useful to use an autoregressive policy network with DDPG?
>
> We consider stochastic policies as the action spaces we consider are discrete and to the extent of our knowledge no successful deterministic method can deal with discrete actions. Indeed, it might be interesting to sequence 1D continuous deterministic actions however as stated in the general response, our aim in this study is not to provide new tools for continuous control. We rather used MuJoCo benchmarks only to show the scalability and correctness of our derivations.
>
> 4) I think that the manual ordering of the sub-action spaces in an auto-regressive policy could introduce bias (i.e. I think there could be a setting that some orderings would make it impossible to learn the optimal policy). Can you comment on this?
>
> Same question as Reviewers 3 and 4, see response to Reviewer 4

---

### Official Review · AnonReviewer4 · 2020-10-28
**Interesting Work | Missing Baselines**

**Rating:** 5
**Confidence:** 3

**Review:**

##########################################################################

Summary:

The paper highlights the difficulty of training with large action space in reinforcement learning. This is usually difficult due to the vast number of possibilities during exploration. They address this issue by studying existing approaches of splitting the action into a finite number of sub-actions and then sampling each sub-action independently or auto-regressively. This leads to a reduction in the candidate actions during exploration which would improve the sample efficiency.  Also, This splitting is referred to as “Factorization of Action Space”.

They majorly study this factorization with PPO and SAC and show how to estimate kl-divergence and entropy in each of these methods. Also, they introduce the notion of using independent q-values for each of the sub-actions which semantically represents expected “sub-action - value” for any action in the current policy with that sub-action.

They show their results on multi-agent and mujoco tasks.

##########################################################################

Reasons for score:

Overall, I do like the paper . However, at this moment, I will keep the paper below acceptance. This is due to the issues raised in Cons section which I believed should be addressed/discussed before I revise my score.

In particular, I would expect authors to have at least  PPO and SAC baseline in their work.

##########################################################################

Pros:

-Large action space has been one of the bottlenecks for scalability in reinforcement learning and the paper studies factorization for on-policy methods which to my understanding has not been done before.

-They provide theoretical analysis for estimation of entropy and kl-divergence in PPO and SAC.

-They show their results on both multi-agent settings as well as control.
##########################################################################

Cons:

- They merely make a comparison between FPPO and FSAC which are the factored versions of respective algorithms and show that FSAC seems to be more robust and has better sample efficiency. HOWEVER, they don’t make comparison with the unfactored ( original) versions of these algorithms. I believe without the comparison with the baseline PPO and SAC, one cannot establish the gain in sample efficiency.

- The number of discrete sub-actions(“m”) for the policy is treated as a hyper-parameter. This is one of the core challenges for continuous control tasks as the environments could be highly sensitive to this choice. They don’t show any concrete evidence for the choice of number of sub-actions made by them, in particular for control tasks.

- How does one decide on the order of the autoregressive actions? This is not clear from their work. Do they manually analyze each sub-action and decide on the order? I believe this order could play a significant role for performance.

##########################################################################

Questions during the rebuttal period:

Please address and clarify the cons above

Optional:
You may want to cite the following:

Q-LEARNING IN ENORMOUS ACTION SPACES VIA AMORTIZED APPROXIMATE MAXIMIZATION (https://arxiv.org/abs/2001.08116).
This paper also talks about auto-regressive actions.


The following series of work involves the decomposition of Q-values for Actor/Critic and Q-learning approaches in multi-agent settings. The major difference is that instead of treating the reward as a vector where each reward component belongs to a specific q-values;  you are giving the same reward to each Q-value.

- Value-Decomposition Networks For Cooperative Multi-Agent Learning
- QMIX: Monotonic Value Function Factorisation for Deep Multi-Agent Reinforcement Learning
- Hybrid Reward Architecture for Reinforcement Learning
- Explainable Reinforcement Learning via Reward Decomposition

##########################################################################

I would be happy to revise my score post clarifications from the authors.

---

> ### Author Response · Authors · 2020-11-24
> **Response to reviewer 4**
>
> We warmly thank the reviewer for the insightful review. The reviewer highlights that:
>
> (a) FPPO and FSAC are not compared with their unfactored versions on MuJoCo environments.
>
>  We agree. However, we would like to clarify that we did not introduce continuous control benchmarks to demonstrate that factored agents outperform their non factored counter-parts, neither in performance nor sample efficiency. Indeed, we observed equivalent performance and sample efficiency for both factored and continuous versions of the agents, with sometimes a slightly improved performance for the continuous version. Our goal was rather to demonstrate the scalability of the approach even when the number of actions becomes very large as well as their correctness and to do it on well-known benchmarks.
>
>
> (b) We do not provide concrete evidence for the choice of the number of sub-actions in the continuous control tasks.
>
> As in Metz et al. 2017, we did not find any strong impact on the performance as long as it is greater than 5. However, we understand the need for a precise study and will add it in a future version of this work.
>
>
> (c) We do not give enough evidence on how the order of the auto-regressive actions is chosen, though it  could play a significant role for performance.
>
> In Platform-v0 this order is straightforward as the agent must first choose the action type and then the action intensity. In Google Football, while we found that enabling auto-regressivity was critical to achieve good performance, we also found that this choice does not impact the results in practice. Finally, we did not use an auto-regressive factorization of the action space for the MuJoCo benchmarks but an independent factorization instead.

---

### Author Response · Authors · 2020-11-24
**General response to all reviewers**

We warmly thank the reviewers for their work and their insightful comments. These reviews helped us realize that this work was not mature enough. We will soon release a stronger version. In particular, we missed several references from the literature and will update the related work accordingly. We also agree that our benchmarks and experiments were not strong enough to highlight the usefulness of factored action spaces.

---

### Decision · Program_Chairs · 2021-01-07
**Final Decision**

**Decision:**

Reject

**Comment:**

This paper studies two techniques for handling high dimensional action spaces in deep RL, namely selecting action components independently or selecting components sequentially in an autoregressive approach.  The methods are developed for two deep RL algorithms (PPO and SAC) and tested on multiple domains.

The reviewers recognized the significance of this research topic but found significant problems in the presentation.  The reviewers raised concerns on the relationship to prior work in the literature (R2), baseline comparisons that are missing in the experiments (R2, R4), and a lack of clarity in the intended message of the experiments (R3).  The authors responded favorably to the reviews, answered clarification questions, and acknowledged the limitations of the submitted work.  The authors expressed their intent to release a stronger paper sometime in the future.  The reviewers acknowledged the author response and were in consensus that the submission needs more work.

Three reviewers have indicated reject for the reasons described above.  The paper is therefore rejected.